# How to Obtain Accurate Environmental Impacts at Early Design Stages in BIM When Using Environmental Product Declaration. A Method to Support Decision-Making

**Elisabetta Palumbo [1,*], Bernardette Soust-Verdaguer [2], Carmen Llatas [2] and Marzia Traverso [1]** 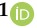

[1] Institute of Sustainability in Civil Engineering (INaB), RWTH Aachen University, D-52074 Aachen, Germany; marzia.traverso@inab.rwth-aachen.de

[2] Instituto Universitario de Arquitectura y Ciencias de la Construcción, Escuela Técnica Superior de Arquitectura, Universidad de Sevilla. Reina Mercedes Avenue 2, 41012 Seville, Spain; bsoust@us.es (B.S.-V.); cllatas@us.es (C.L.)

[*] Correspondence: elisabetta.palumbo@inab.rwth-aachen.de

**Abstract:** The construction sector plays an important role in moving towards a low-carbon economy. Life cycle assessment (LCA) is considered one of the most effective methods of analytically evaluating environmental profiles and an efficient tool for calculating the environmental impacts in building design-oriented methodologies, such as building information modelling (BIM). At early design stages, generic LCA databases are used to conduct the life cycle inventory (LCI), while detailed stages require more detailed data, such as environmental product declarations (EPDs), namely documents that provide accurate results and precise analyses based on LCA. Limitations are recognized when using EPDs in BIM elements at different levels of development (LOD) in the design stages, especially related to the data consistency and system boundaries of the LCA. This paper presents a method of achieving accurate LCA results, that helps with decision-making and provides support in the selection of building products and materials. The method is validated by its application in the structural concrete of an office building located in Germany. The method defines a safety factor adopted for embodied impacts ("cradle-to-gate"), based on EPD results to predict the environmental impact of BIM elements at different LODs. The results obtained show that by integrating the method to conduct the LCA, the range of errors and possible inconsistencies in the LCA results can be reduced.

**Keywords:** LCA (life cycle assessment); environmental product declaration (EPD); BIM (building information modelling); LOD (level of development); building sustainability; from cradle to gate; early design stages

---

## 1. Introduction

### 1.1. Context

Considering the current environmental impacts, the relevance of reducing the environmental impact of buildings is becoming urgent. Decreasing the environmental, economic and social impacts is a primary objective of the European Union (EU), which aims to develop a sustainable, competitive, safe and decarbonised energy system [1]. The Sustainable Development Goals presented in the UN Agenda 2030 [2] promote taking urgent action to protect the climate and make cities and human settlements inclusive, safe, resilient and sustainable. One of the main challenges to address is the reduction in the carbon emissions of our buildings [2].

The construction sector is responsible for a high global share of energy consumption (42%), greenhouse gas emissions (35%), over 50% of the extraction of materials, 30% of water withdrawal and approximately one-third of the waste, and consequently it plays an important role in moving towards the transformation of the European economy into a more sustainable economy by 2050 [3]. Given that the building sector has a considerable influence on the move towards a more sustainably built environment (and is subsequently largely responsible for facilitating this movement), all the stakeholders involved in the supply chain, including architects, designers, engineers and other players involved in the construction sector, are in a unique position that enables them to set sustainability objectives to reduce the environmental impact from the early stage of a building project [4]. Robust tools and accurate information are required to help them, from the very early design stages to assess and consider the environmental impacts in projects and building constructions.

### 1.2. Challenges in the Life Cycle Assessment Application in the Building Sector

The life cycle assessment (LCA) is considered the most appropriate scientific methodology to assess the environmental impacts within the entire building life cycle [5], as it provides analytical, reliable and life-cycle-spanning quantifications [6] e.g., embodied energy (EE), operational energy (OE) and global warming potential (GWP) effects [7], among others. A recent study [8] reports that almost all LCA studies applied to the building sector, conducted in the design phase, did not influence the design decisions. The reason for this probably lies in the fact that the LCA method requires large amounts of data and calculations, and the amount of information to collect and compile is vast. Therefore, a detailed LCA is often conducted after the building has been built and all the information is available [9]. Whereas if applied at the initial design stage, the evaluation and improvement of environmental performance represent a highly effective action to direct buildings towards a careful use of resources and impact reduction, given that the decisions made at this stage are relevant and wide-ranging [10–14]. In this context, professionals, decision-makers and investors throughout the EU require empirical, reliable, transparent and comparable data [15], in turn based on clear building performance indicators.

### 1.3. The Problem of Consistency in LCA Data Sources

The Environmental Product Declaration (EPD), also called type III environmental declaration in ISO 14020:2000 [16], standardised by ISO 14025:2006 [17], is LCA-based information that provides reliable and verifiable information on a specific product [18]. The EN 15804:2012+A2:2019 [19] standard proposed a procedure for calculating the environmental impact of construction products and defined five different types of EPD depending on the LCA system boundaries (Figure 1): (1) cradle to gate (A1–A3); (2) cradle to gate with mandatory C1–C4 and D; (3) cradle to gate with options (C1–C4 and D); (4) cradle to gate with options (A4 and A5); and (5) cradle to grave with mandatory D.

The European Standard 15978 [20] standard, the adaptation of buildings to the LCA method, proposes a calculation procedure based on the addition of the resulting bill of material/product quantities and process quantities and the environmental impacts of products and processes in the different life cycle stages of the building. This means that, for example, for the GWP calculation, the resulting impact is the following equation:

$$\begin{aligned} \text{GWP Building} = \ & \text{GWP Product Stage } (A1 - A3) + \text{GWP Construction Stage } (A4 - A5) \\ & + \text{GWP Use Stage } (B1 - B7) + \text{GWP End Of Life Stage } (C1 - C4) \end{aligned} \tag{1}$$

where GWP i LC stage $=$ Amount of Product or Process $(a_1)$ $\times$ GWP per unit of Product or Process.

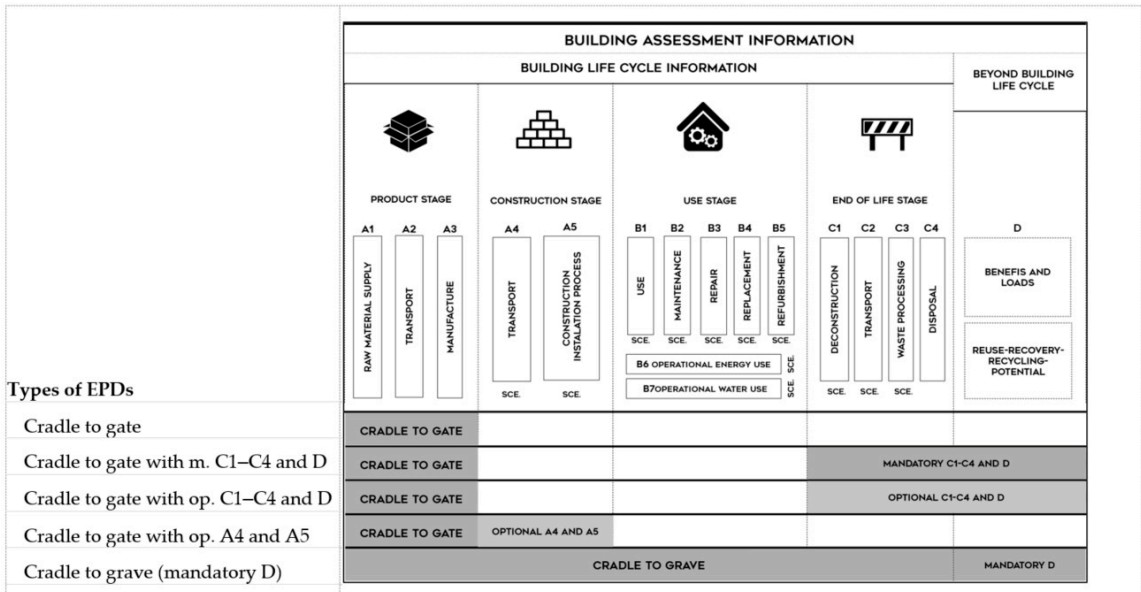

**Figure 1.** Building assessment information modules and types of EPDs according to EN15804:2012+A2:2019 [19] standard.

Figure 2 shows that according to the EN 15978 [20] standard, the basic principle of the matrix calculation routine consists of multiplying each product and service quantified (bill of quantities) in a certain module of the life cycle of the building by its respective environmental indicator value per unit of products/processes.

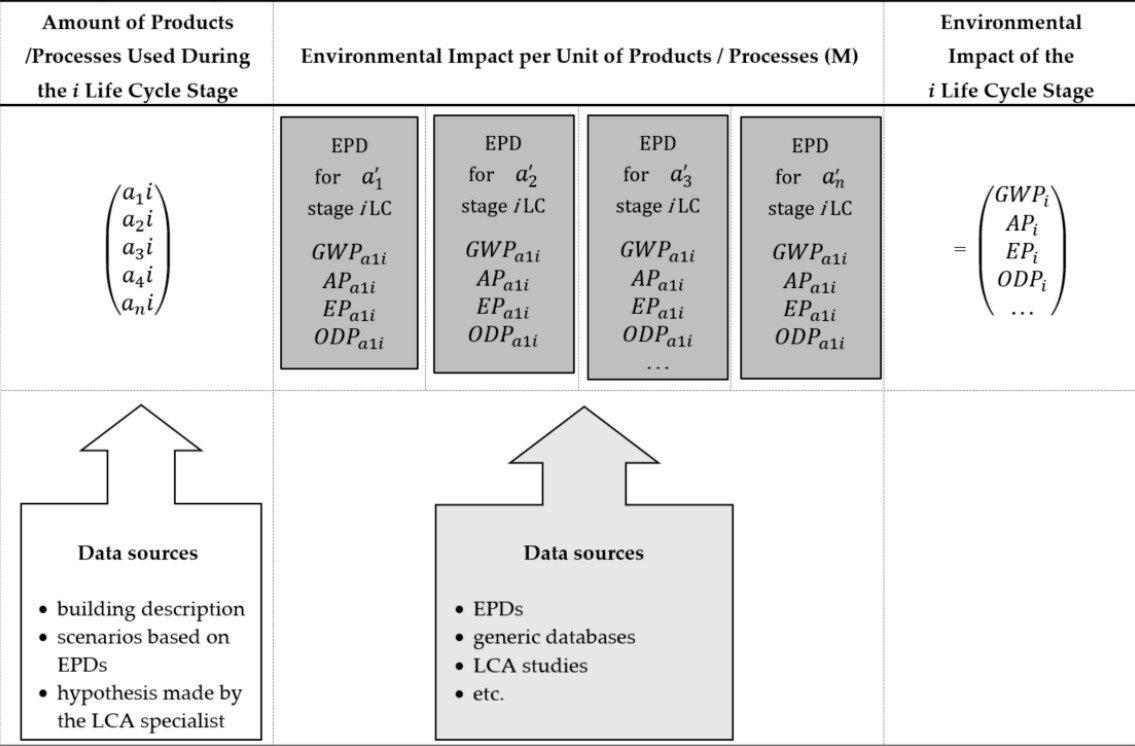

**Figure 2.** Calculation principle based on EN 15978 [20] standard.

The data sources to extract the environmental indicators per unit of product/process can be generic (e.g., generic LCA databases) or specific (e.g., environmental product declaration(EPD)).

The generic databases are mainly provided by academics and consultancy firms, (e.g., Ecoinvent centre, PE International, and European Commission, Joint Research Centre) and specific data are provided by the industry sector and based on specific products (e.g., EPD specific programs) [21]. The use of generic data can be used to describe environmental impacts in a national or regional context but not for specific products [22].

The use of specific rather than generic data, recommended by the International Reference Life Cycle Data System (ILCD) guide [22], can produce more reliable and accurate results, but this is not always possible when there is no existing EPD for a certain product and when uncertainties are surrounding the use of specific products. The use of EPDs to assess the environmental performance of buildings is becoming increasingly important [23]. This provides evidence of the relevance of using EPD in the application of an LCA to an entire building which we consider as the basis for this study.

Otherwise, the existence of different data sources (which means different data granularity (generic or specific) and different data sources (e.g., Ecoinvent, Gabi, etc.) to conduct an LCA would mean that data consistency cannot always be ensured when combining them. This problem was previously addressed in the literature [21,23–25]. Emami et al. [25], analysing two different types of residential buildings (a concrete-element multi-story residential building and a detached wooden house) and adopting the two most widely used life cycle assessment (LCA) database–software combinations, Ecoinvent with SimaPro software and GaBi, concluded that the different results obtained for the building systems (different parts of the buildings) and material levels were delivered in almost all impact categories. The study investigates the impact of using the assessment tool on the results of the assessment of the pre-use phase emissions of two recently built typical residential buildings in Finland. By analysing the results, one of the possible causes for the differences observed between the two database-software combinations was the reference country and the energy mix of said reference country. Another study [23] focuses its efforts on comparing the current situation in environmental construction product programs in different European countries (including Austria, Belgium, France, Germany, Switzerland), and shows that further harmonization across the different environmental product declarations (including EPDs, PEFs and the CPR (BR 710)) is needed to avoid the current confusion.

In the comparison of the environmental impacts of the outer envelope of an Italian passive house, calculated by using different data sources, EPDs and the active house LCA evaluation tool (AH Ltool) with different granularity (element and material levels), Palumbo et al. [26] demonstrate the limitations of generic data sources such as the AH Ltool compared with the EPD (based on specific manufacturers). The study concluded that data from the EPD and AH Ltool differ from each other in several impact indicators. Moreover, Lasvaux et al. [21], who compared generic and product-specific LCA databases, also concluded that current generic and EPD databases presented very different values at the database scale which depend on the type of environmental indicator. This means that some indicators such as fossil fuel consumption are less variable than indicators that require a higher number of elementary flows such as photochemical oxidation (POCP). This reinforces the statement that there are still gaps to fulfil to ensure data consistency in LCA studies, especially considering the harmonization of data with different granularity or levels of detail (such as generic databases and EPD).

### 1.4. Challenges in Data Sources of BIM-Based LCA in Building Design Phases

In recent decades, the use of the building information modelling (BIM) methodology in the construction sector has led to significant changes with respect to design, construction, management and commissioning processes [22]. Thus, BIM is considered an efficient support to the automated application of LCA in the building design stages [27].

The advantages of integrating BIM methodology and LCA have been recognized in the literature [28] as a simplification strategy that can reduce additional effort and time in the LCA application. The use of BIM to conduct LCA can, for example, speed up the extraction of the bill of material quantities [29–31]. Existing reviews [30,32] show growing interest in the field, which

is revealed, inter alia, by the increasing number of publications. Among existing BIM-based LCA studies, the comparison of different data sources at early and detailed stages of design in BIM has been addressed. Shadram et al. [33] proposed a method to deal with different EPD in BIM-based LCA. The method automatically compares different types of EPDs. However, although different alternatives during the detailed stage of design are compared, the study does not consider alternatives to deal with data in the early design stages. Bueno and Fabricio [34] compared the results of different data sources integrated into design tools: a full LCA on Gabi 6 software and a BIM-LCA plug-in. The study concluded that no consistent results can be assured. Rezaei et al. [9] propose a method of conducting LCA that considers the different design stages and LOD of the BIM model, but based on a generic database (Ecoinvent 3.3) [35] and used to allocate the appropriated process considering both early and detail design stages.

The problem of data granularity in conducting BIM-based LCA was also addressed by Cavalliere et al. [36], by using BIM and mixing LCA databases with different levels of detail. The method divides the building into functional elements and construction categories and matches the LOD of these elements with different LCA databases, but does not propose a strategy to deal with information at a material or product level (such as EPD). Santos et al. [37] proposed an approach for dealing with (environmental and economic) enriched BIM objects to conduct LCA in BIM considering early and detailed design stages (streamlined and complete LCA) and, like Durão et al. [38], they proposed the use of generic environmental information (e.g., generic databases) for lower LODs and more specific information (e.g., EPD) for higher LODs. In theory, this is a suitable way to include EPD indicators in the BIM-based LCA, however, there may be no correspondence or consistency between the LCA results and the LOD of the BIM model in the design stages.

## 1.5. Challenges in EPD Integration in BIM

The reviewed literature confirms that several weakness and unsolved issues were detected when integrating data sources (EPD) in BIM-based LCA:

- Considering the different types of EPDs (arising from the LCA system boundaries), there were limitations in adapting the impacts produced in modules A4 to D of the current EPD format to the building's specific characteristics. This means that, concerning the environmental indicators for modules of information from A4 to D, the scenarios and assumptions are closely linked to specific scenarios and assumptions adopted in the EPD, and therefore cannot be used directly. For example, in the transport module (A4), the distance from the manufacturing site to the building site declared in the in EPD can be 25 km, (adopting a vehicle with emission standards Euro 4 and capacity of 30 tons), and then when using the EPD for a specific LCA case study, the distances and lorry capacity can differ from the information assumed in the EPD (which can be for example 200 km with Euro 3 and a 15 ton load). This fact can produce a broad margin of errors when transforming the values taken from the EPD and adapting them to a case study. Thus, when calculating the total impacts as the addition of all the EPDs and all information modules, as suggested in EN 15978 [20], without adapting the EPD results for modules A4 to D to the specific characteristics of the buildings, the use of the EPD "cradle-to-gate" type in the design stages can accumulate fewer errors. A possible solution could be to limit the use of EPD to the range of impacts produced in the product stage ("cradle-to-gate"), avoiding possible errors or double counting of A4 to D modules when conducting LCA in the design process in BIM.
- Lack of consistency between different data granularity (generic versus specific) in the building design process in BIM. In fact, in the low LODs, at early design stages, the specific materials to be used are as yet unknown, and consequently to solve this several references [21,37] perform an LCA study with generic LCA data at the early design stages (up to LOD 300) and data from EPDs in the detailed stages (from LOD 400). However, what if when comparing BIM-based LCA early design results with BIM-based LCA detailed design results, there is no data consistency related to

the use of different data granularity (e.g., generic database or building environmental catalogue and EPDs)?

A possible solution could be to determine an accurate range of generic values used at the early design stages in keeping with specific values at the detailed design stages in BIM. Thus, as it is not possible to use the EPDs of specific products in the early design stages, an effective way to proceed would be to adopt a default "safety" factor. A similar approach has been adopted by the German Sustainable Building Council (*Deutsche Gesellschaft für Nachhaltiges Bauen—DGNB*) system [39] by introducing a safety factor (10%), added in case the implemented EPDs do not exactly match the materials and components of the building. Currently, the authors have not identified BIM-based LCA publications focused on methods to harmonize the use of EPDs in the design stages in BIM, considering early and detailed design stages.

To bridge these gaps in the existing literature, by comparing different products for a building office structure, here we propose a method to deal with different data granularity and reduce unpredictable differences or data inconsistency in the use of the EPD in the different design stages in BIM. The case study application also illustrates the limitations of current the EPD scheme to conduct a whole life cycle assessment of building products according to the building design stages in BIM methodology. However, this work will not examine the operative integration of EPD modules in the BIM model.

## 2. Materials and Method

### 2.1. A Method for Using EPD in BIM-Based LCA

Given that EPDs are normally used in BIM-based LCA application when the elements of the BIM model are mostly around LOD 300 [37] or higher and the literature shows that EPD is a preferable LCA source [22], in this paper we develop a method that enables us to obtain consistent and accurate LCA results, dealing with the use of EPDs at different design stages in BIM.

The proposed method consists of defining the range values (minimum, maximum and median) of the embodied impacts of building material (e.g., concrete) extracted from the EPD certification. Based on the characterization of that material (e.g., concrete), it can be used at the early design stage when the commercial name of the product, code and manufacturer are unknown and it is not possible to choose a specific EPD. This range of values can also form the basis for defining a safety factor adopted in the early design stage.

The decision to base the LCA indicators of a building material on the EPD of manufacturers in early design stages in BIM, instead of generic datasets based on LCA, aims to obtain accurate results between the early and the detailed design stages in BIM, when a specific EPD is normally used.

To address these challenges, starting with an office building, we have focused the attention on the definition of environmental impacts based on EPD sources concerning a specific material from the building structure: concrete. Concrete was chosen, as the literature [40,41] considers it a building material with one of its main priorities (in magnitude and number of sources) being to reduce a building's environmental impacts in the product stage.

Starting from the LCA outcomes related to the structure, this study discusses whether EPDs can define an accurate LCA of structural concrete in the early stages, by adopting a safety factor in the low LOD when the characteristics and producer of the material are not completely known. The safety factor is a predetermined factor that should be used instead of generic data to ensure more reliable and accurate results in the early design stage, given that there are uncertainties surrounding the use of specific products and, consequently, it is not possible to choose specific EPDs.

The definition of a safety factor is determined through the EPDs collected in the dedicated platforms in the final restricted step on the Institut Bauen und Umwelt (IBU) platform [42], on the grounds of comparability.

Thereby, the results of the case study at the detailed design stage are compared with the environmental impacts of the building with the same characteristics in the early design stage

(mostly LOD 100), obtained considering environmental impact categories taken from EPDs. However, in this study, we assume a similar amount of structural concrete for both the early and detailed design stages, which is justified by consideration of the low variability of the structure system in the design stages in BIM [31].

In order to examine this, the work has been broken down into 4 steps (Figure 3):

- Step 1: determination of the environmental profile of the selected material (concrete) considering a case study;
- Step 2: collection of EPDs for products made of the selected material (concrete) and selection of materials suitable for structural uses;
- Step 3: classification of the EPD indicators on a performance basis and in compliance with EPD comparability requirements defined by ISO 14025 [17];
- Step 4: determination of the safety factor and range factor (minimum, maximum, average, median) of EPD indicator values from the early to detailed design stages and integration in BIM.

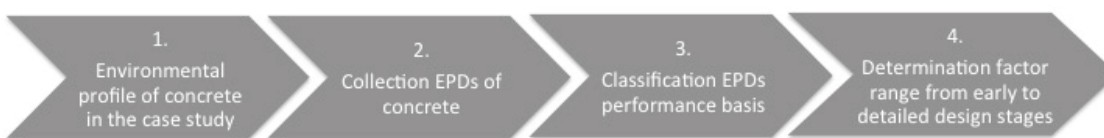

**Figure 3.** Flowchart of the main steps.

## 2.2. Case Study: Concrete Structure of an Office Building

The proposed method is validated through its application in a case study. The case study verification was based on the structure of an office building hypothetically located in Germany. The building is made up of laboratories, offices, and common areas. The building is a four-story, rectangular shape with a useful area of 19,307 m$^2$ and 14 m high. The primary energy consumption of the building conforms with class A at national scale, with a winter thermal index of less than 30 kWh/m$^2$ per year.

The building has a load-bearing structure consisting of a frame of beams and pillars made of concrete, modulated on a structural mesh measuring $605 \times 500$ cm and $605 \times 800$ cm, which is the basic module for the façades, with minimum dimensions of $302.5 \times 500$ cm and $302.5 \times 800$ cm. Two parts of the building have been analysed: the foundation ("substructure") and the horizontal and vertical structure ("substructure"). The foundation, consisting of piles, pile caps and columns, is made up of 11,410.85 m$^3$ of concrete and 604,408 kg of steel bars, while the superstructure is made up of 6321.76 m$^3$ of concrete and 1,498,024 kg of steel bars (Figure 4). Figure 5 shows the three-dimensional (3D) model of the building.

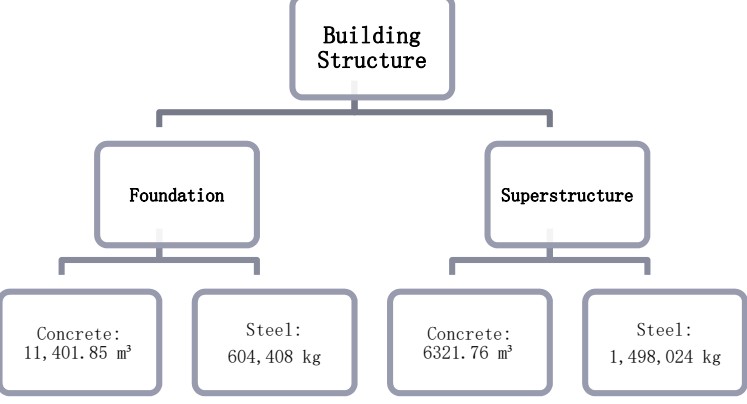

**Figure 4.** Bill of material relating to the structure.

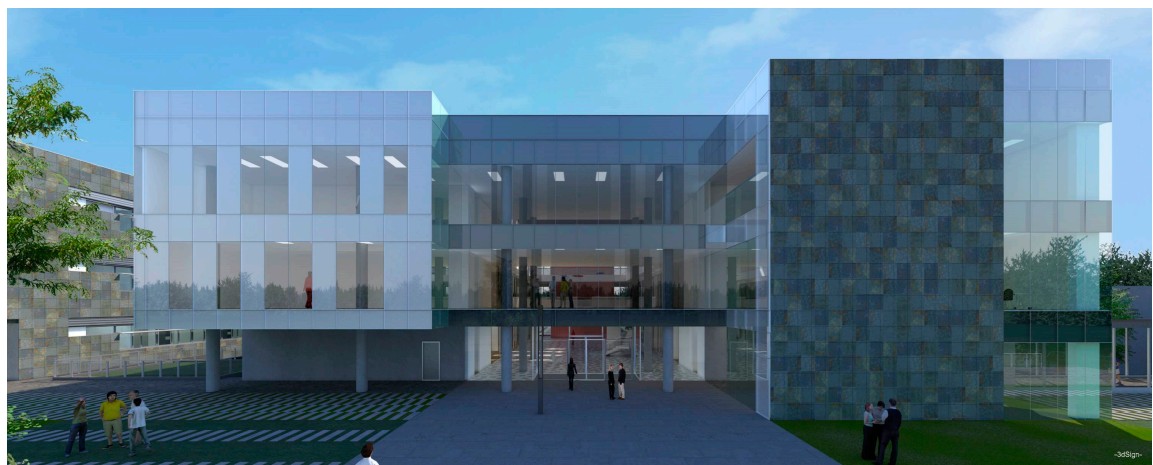

**Figure 5.** 3D model of the building (image by Politecnica Ingegneria & Architettura [17]).

2.2.1. Step 1: Environmental Profile of the Selected Material (Concrete)

The definition of the structure's environmental profile was defined by adopting an EPD of 1 m$^3$ of concrete, the density of 2400 kg/m$^3$ and compressive strength class C 25/30 [43] according to German Institute for Standardization (*Deutsches Institut für Normung eV*) DIN EN 206-1 [44]. The EPD adopted concerns a concrete certified by InformationsZentrum Beton GmbH.

In accordance with standard EN 15804 [19], the main LCA results are grouped into 3 categories: environmental indicators (e.g., GWP, AP, etc), resource use (e.g., use of renewable primary energy resources used as raw materials; use of renewable secondary fuels, etc.) and output flows and waste categories. With the intention of limiting the scope of this study and the case study verification, not all the EPD results have been used, and instead the impact categories reported in Table 1 are illustrated.

**Table 1.** Analysed impact categories.

| Impact Category | Impact Category Abbreviation | Unit |
|---|---|---|
| Global Warming Potential | GWP | kg $CO_2$ eq. |
| Ozone layer depletion | ODP | kg CFC 11eq. |
| Acidification Potential | AP | kg $SO_2$ eq. |
| Eutrophication Potential | EP | kg $PO_4$ eq. |
| Photochemical oxidation | POCP | kg Ethene eq. |
| Abiotic depletion potential resources—elements | ADPe | kg Sb eq. |
| Abiotic depletion potential resources—fossil resources | ADPf | MJ |

Table 2 illustrates the distribution of the environmental impacts for the selected indicators (GWP, ozone layer depletion (ODP), acidification potential (AP), eutrophication potential (EP), POCP, abiotic depletion potential resources—elements (ADPe)) referring to the manufacturing process (A1–A3 stages) of 1 m$^3$ considering the EPD defined above [43], and the total amount of concrete considered in the case study (17,732.61 m$^3$), respectively.

**Table 2.** Environmental impacts of concrete structure at an estimated level of development (LOD 350) (A1–A3).

| Functional Unit | Reference | Environmental Impact | | | | | | |
|---|---|---|---|---|---|---|---|---|
| | | GWP | ODP | AP | EP | POCP | ADPe | ADPf |
| 1 m$^3$ | [43] | $1.97 \times 10^2$ | $5.36 \times 10^{-8}$ | $2.87 \times 10^{-1}$ | $5.35 \times 10^{-2}$ | $2.30 \times 10^{-2}$ | $6.43 \times 10^{-4}$ | $9.00 \times 10^2$ |
| 17,732.61 m$^3$ | [45] | $3.49 \times 10^6$ | $9.50 \times 10^{-4}$ | $5.09 \times 10^3$ | $9.49 \times 10^2$ | $4.08 \times 10^2$ | $1.14 \times 10^1$ | $1.60 \times 10^7$ |

2.2.2. Step 2: Collection of EPDs

In this step, the EPDs presented in the main program operators (POs) relating to the building sector were selected. The analysed POs were respectively IBU (Germany), EPD Norge (Norway), EPD Italy (Italy), the EPD International AB (Sweden), the independent international certification body Building Research Establishment BRE Global (United Kingdom) and INIES (France).

The following have been excluded from this initial analysis:

- Labels drawn up in a language other than English or German (e.g., in EPD Norge or in EPD Italy most of them are in the national language);
- Those referring to products whose geographic context is outside of Europe (e.g., 7 of those belonging to the Swedish PO referred to countries such as India, Brazil, the United Arab Emirates, Chile and Australia);
- Those concerning productions or average products and non-specific products.

Subsequently, the EPDs were organized considering technical data such as density, the compressive strength, the composition (base and ancillary materials) of products and the suitability of products covered by EPDs for structural uses in an office building.

Specific data are available for fourteen materials (Table 3). Available EPDs from the current market are mostly the "cradle to gate" type (EN 15804), therefore, the LCA is limited to impacts from stages A1 to A3. They all declare the mandatory production model (A1–A3), whereas only 60% declare the "benefits and loads beyond the system boundaries" module (D), and 70% consider the "use" and "end of life" modules. This can be due to the fact that, beyond the production phases, the data needed to include the impact produced during the specific building construction process (A4–A5), its operation (B1–B7) and its end of life (C1–D) have not yet been defined, thus, the high variability and uncertainty prevents reliable results.

Therefore, given the lack of available data, the study is limited to the impacts produced during the production phase (module A1–A3). As mentioned above, to limit the study, only the impact categories reported in Table 2 are shown.

The values of the six considered impacts categories (GWP, ODP, AP, EP, POCP, ADPe) concerning the selected EPDs in the production phase (A1–A3) are reported with reference to the functional unit, i.e., 1 m$^3$ of concrete (Table 4).

A first comparison of the EPD results showed substantial deviations between products with similar compressive strengths. For example, for the group of products with compressive strength C35–45 (EPD 3–6 and 8) the percentage difference in GWP and POCP is about 60%, about 86% in AP and well above 100% in the rest of the indicators. Even within the broader "ordinary concrete" range (compressive strength from C20/25 to C55) there is a large percentage variation (Table 5).

This can be related to the different and non-comparable product category rules (PCR) adopted, and then different LCA assumptions. Product category rules, regulated by ISO 14025 [17], are additional rules on how to conduct an LCA and other types of sustainability information for each product category.

Accordingly, in order to promote fair comparability between the selected EPDs, another restriction of the field was made, focusing attention on the labels obtained from the same PCR and therefore recorded in the same program operator. In fact, compliance with the requirements established by the ISO standard (covering the same product category, LCA methodology, environmental indicators, additional environmental information, life cycle stages covered, EPD validity, etc., see 6.7.2 ISO 14025 [17]) can be achieved most easily by two EPDs based on the same product category rules and developed based on the general LCA methodology in the same EPD programme [46,47].

**Table 3.** Overview of EPDs for concrete analysed and collected.

| EPD | Name | Owner | Validity (Year) | Country | Density (kg/m$^3$) | Compressive Strength (N/mm$^2$) | Programme Operator |
|---|---|---|---|---|---|---|---|
| EPD 1 | AgillaTM Ready-mix Concrete Aggregate Industries UK Limited | BASF | 2022 | UK | 2332 | 40 | IBU |
| EPD 2 | Generic ready-mixed concrete | BRMCA | 2023 | UK | 2380 | 30/37 | IBU |
| EPD 3 | Beton der Druckfestigkeitsklasse C 35/45 | Informations Zentrum Beton GmbH | 2023 | DE | 2400 | 35/45 | IBU |
| EPD 4 | Beton der Druckfestigkeitsklasse C 30/37 | Informations Zentrum Beton GmbH | 2023 | DE | 2400 | 30/37 | IBU |
| EPD 5 | Ready-mix concrete | General Beton Romania | 2022 | RO | 2300 | 20/25 | EPD International AB |
| EPD 6 | Ready-mix concrete | General Beton Romania | 2022 | RO | 2300 | 25/30 | EPD International AB |
| EPD 6 | Ready-mix concrete | General Beton Romania | 2022 | RO | 2300 | 30/37 | EPD International AB |
| EPD 6 | Ready-mix concrete | General Beton Romania | 2022 | RO | 2300 | 35/45 | EPD International AB |
| EPD 6 | Ready-mix concrete | General Beton Romania | 2022 | RO | 2300 | 40/50 | EPD International AB |
| EPD 7 | C32/40 CEMI Ready Mix Concrete | Hanson UK | 2023 | UK | 2000–2600 | 40 | Bre global |
| EPD 8 | C28/35 CIIIA Ready Mix Concrete | Hanson UK | 2023 | UK | 2000–2600 | 35 | Bre global |
| EPD 9 | Beton der Druckfestigkeitsklasse C 25/30 InformationsZentrum Beton GmbH | InformationsZentrum Beton GmbH | 2023 | DE | 2400 | 25/30 | IBU |
| EPD 10 | Beton der Druckfestigkeitsklasse C 20/25 Informations Zentrum Beton GmbH | InformationsZentrum Beton GmbH | 2023 | DE | 2400 | 20/25 | IBU |
| EPD 11 | Beton der Druckfestigkeitsklasse C 45/55 | InformationsZentrum Beton GmbH | 2023 | DE | 2400 | 45/55 | IBU |
| EPD 12 | WatertightTM Ready-mixed Concrete | BASF plc, UK limited | 2021 | UK | 2386 | 50 | IBU |
| EPD 13 | Beton der Druckfestigkeitsklasse C 50/60 | InformationsZentrum Beton GmbH | 2023 | DE | 2400 | 50/60 | IBU |
| EPD 14 | DiamondconcreteTM Ready-mixed Concrete | BASF plc, UK limited | 2021 | UK | 2385 | 85 | IBU |

**Table 4.** Results of LCA of 1 m$^3$ of concrete, based on the collected EPDs from cradle to gate (stages A1–A3).

| EPD | Compressive Strength | GWP | ODP | AP | EP | POCP | ADPe | ADPf |
|---|---|---|---|---|---|---|---|---|
| N° | (N/mm$^2$) | kg CO$_2$ eq. | kg CFC 11eq. | kg SO$_2$ eq. | kg PO$_4$ eq. | kg Ethene eq. | kg Sb eq. | MJ |
| EPD1 | 40 | $3.73 \times 10^2$ | $1.72 \times 10^{-8}$ | $9.02 \times 10^{-1}$ | $1.22 \times 10^{-1}$ | $8.84 \times 10^{-2}$ | $3.51 \times 10^{-4}$ | $2.23 \times 10^3$ |
| EPD2 | 30/37 | $2.20 \times 10^2$ | $1.69 \times 10^{-6}$ | $3.47 \times 10^{-2}$ | $3.47 \times 10^{-2}$ | $1.16 \times 10^{-1}$ | $4.40 \times 10^{-4}$ | $1.12 \times 10^2$ |
| EPD3 | 35/45 | $2.44 \times 10^2$ | $6.81 \times 10^{-8}$ | $3.48 \times 10^{-1}$ | $6.55 \times 10^{-2}$ | $2.98 \times 10^{-2}$ | $8.22 \times 10^{-4}$ | $1.08 \times 10^3$ |
| EPD4 | 30/37 | $2.19 \times 10^2$ | $5.97 \times 10^{-8}$ | $3.17 \times 10^{-1}$ | $5.91 \times 10^{-2}$ | $2.58 \times 10^{-2}$ | $7.11 \times 10^{-4}$ | $9.97 \times 10^2$ |
| EPD5 | 20/25 | $3.01 \times 10^2$ | $1.11 \times 10^{-5}$ | $7.00 \times 10^{-1}$ | $2.15 \times 10^{-1}$ | $3.00 \times 10^{-2}$ | $2.44 \times 10^{-4}$ | $1.50 \times 10^3$ |
| EPD6 | 25/30 | $3.10 \times 10^2$ | $1.14 \times 10^{-5}$ | $7.15 \times 10^{-1}$ | $2.20 \times 10^{-1}$ | $3.00 \times 10^{-2}$ | $2.49 \times 10^{-4}$ | $1.54 \times 10^3$ |
| EPD6 | 30/37 | $3.41 \times 10^2$ | $1.22 \times 10^{-5}$ | $7.80 \times 10^{-1}$ | $2.40 \times 10^{-1}$ | $3.00 \times 10^{-2}$ | $2.66 \times 10^{-4}$ | $1.68 \times 10^3$ |
| EPD6 | 35/45 | $3.85 \times 10^2$ | $1.35 \times 10^{-5}$ | $8.70 \times 10^{-1}$ | $2.60 \times 10^{-1}$ | $3.00 \times 10^{-2}$ | $2.90 \times 10^{-4}$ | $1.90 \times 10^3$ |
| EPD6 | 40/50 | $4.10 \times 10^2$ | $1.41 \times 10^{-5}$ | $9.20 \times 10^{-1}$ | $2.80 \times 10^{-1}$ | $4.00 \times 10^{-2}$ | $2.98 \times 10^{-4}$ | $1.92 \times 10^3$ |
| EPD7 | 40 | $3.71 \times 10^2$ | $8.38 \times 10^{-6}$ | $7.23 \times 10^{-1}$ | $2.45 \times 10^{-1}$ | $7.53 \times 10^{-2}$ | $1.22 \times 10^{-4}$ | $2.01 \times 10^3$ |
| EPD8 | 35 | $2.13 \times 10^2$ | $7.68 \times 10^{-6}$ | $5.04 \times 10^{-1}$ | $1.57 \times 10^{-1}$ | $5.65 \times 10^{-2}$ | $9.58 \times 10^{-5}$ | $1.43 \times 10^3$ |
| EPD9 | 25/30 | $1.97 \times 10^2$ | $5.36 \times 10^{-8}$ | $2.87 \times 10^{-1}$ | $5.35 \times 10^{-2}$ | $2.30 \times 10^{-2}$ | $6.43 \times 10^{-4}$ | $9.00 \times 10^2$ |
| EPD10 | 20/25 | $1.78 \times 10^2$ | $4.79 \times 10^{-8}$ | $2.61 \times 10^{-1}$ | $4.98 \times 10^{-2}$ | $2.05 \times 10^{-2}$ | $6.08 \times 10^{-4}$ | $8.19 \times 10^2$ |
| EPD 11 | 45/55 | $2.86 \times 10^2$ | $7.72 \times 10^{-8}$ | $4.06 \times 10^{-1}$ | $8.10 \times 10^{-2}$ | $3.51 \times 10^{-2}$ | $1.02 \times 10^{-3}$ | $1.36 \times 10^2$ |
| EPD 12 | 50 | $2.54 \times 10^2$ | $1.17 \times 10^{-8}$ | $6.72 \times 10^{-1}$ | $8.83 \times 10^{-2}$ | $6.46 \times 10^{-2}$ | $2.20 \times 10^{-4}$ | $1.57 \times 10^3$ |
| EPD 13 | 50/60 | $3.00 \times 10^2$ | $8.40 \times 10^{-8}$ | $4.22 \times 10^{-1}$ | $8.37 \times 10^{-2}$ | $3.79 \times 10^{-2}$ | $1.07 \times 10^{-3}$ | $1.36 \times 10^3$ |
| EPD 14 | 85 | $4.66 \times 10^2$ | $9.38 \times 10^{-8}$ | $1.01$ | $1.40 \times 10^{-1}$ | $8.42 \times 10^{-2}$ | $5.08 \times 10^{-4}$ | $2.23 \times 10^3$ |

**Table 5.** Percentage of difference of the impact indicators from Compressive strength class 20 to class 55.

| | Impact Indicators | | | | | | |
|---|---|---|---|---|---|---|---|
| | GWP | ODP | AP | EP | POCP | ADPe | ADPf |
| C25/30 vs. C20/25 | 3.1 ÷ 10.7% | 2.7 ÷ 11.9% | 2.1 ÷ 10% | 2.3 ÷ 7.4% | 0.0 ÷ 12.2% | 2.1 ÷ 5.8% | 3.0 ÷ 9.9% |
| C30/37 vs. C25/30 | 9.9 ÷ 11.7% | 7.5 ÷ >100% | −87.9% ÷ 9.1 | −35.1 ÷ 9.1 | 0.0 ÷ >100% | −31.6% ÷ 7.0 | −87.6% ÷ 8.7 |
| C35/45 vs. C30/37 | 10.9 ÷ 12.9% | −96 ÷ 10.7% | 11.5% ÷ >100% | −74.3 ÷ −8.3% | 0.0 ÷ >100% | 7.0 ÷ −31.6% | 8.7 ÷ −87.6% |
| C40/50 vs. C35/45 | −3.6 ÷ 6.4% | −82.8 ÷ 4.4% | −16.9 ÷ 93.1% | −5.8 ÷ 34.8% | >100% ÷ 33.3 | −73.2 ÷ 2.8 | 0.9 ÷ 45.4% |
| C45/55 vs. C40/45 | −30.2 ÷ 12.6% | −99.5 ÷ >100% | −55.9 ÷ −39.6% | −66.9 ÷ −8.3% | −53.4 ÷ −12.3% | >100% | −32.3 ÷ −13.4% |

Therefore, considering the geographic context of the building studied, the selection was concentrated on the labels recorded in the German Institut Bauen und Umwelt e.V. system [42].

The principles that drove the selection of EPDs in IBU were: the adoption of PCR 'Concrete components made of in-situ or ready-mixed concrete' [48] and an effective validity check of the label within 1 year, which was considered the time necessary to conduct our assessment. Consequently, they were systematized based on the validity date, geographical representation, density, functional unit and compressive strength (Table 6).

**Table 6.** Overview of EPDs related to the concrete product present in the IBU system.

| N° | Reference for EPD | Date of Validity | Country | Density | Compressive Strength | Functional Unit |
|---|---|---|---|---|---|---|
| EPD1 | [49] | 2023 | DE | 2400 | 20/25 | 1m³ |
| EPD2 | [43] | 2023 | DE | 2400 | 25/30 | 1m³ |
| EPD3 | [50] | 2023 | UK | 2380 | 30/37 | 1m³ |
| EPD4 | [51] | 2023 | DE | 2400 | 30/37 | 1m³ |
| EPD5 | [52] | 2023 | DE | 2400 | 35/45 | 1m³ |
| EPD6 | [53] | 2022 | UK | 2332 | 40 | 1m³ |
| EPD7 | [54] | 2023 | DE | 2400 | 45/55 | 1m³ |
| EPD8 | [55] | 2021 | UK | 2386 | 50 | 1m³ |
| EPD9 | [56] | 2023 | DE | 2400 | 50/60 | 1m³ |
| EPD10 | [57] | 2021 | UK | 2385 | 85 | 1m³ |

### 2.2.3. Step 3: Classification of EPDs Based on Structural Performance

The second step of the work dealt with the collection of characterized indicators retrieved from the Environmental Product Declarations of the concrete manufacturers registered within the International IBU System.

Furthermore, in order to define (minimum and maximum) reference values for concrete products on a performance basis, a classification of concrete based on technical characteristics was identified, taking into account the standards DIN EN 206:2017-1 [44] and DIN 1045-2:2008 [58].

These standards divide the concretes, on the basis of the values of the characteristic compressive strength, into the following groups:

- Low strength concrete (LSC): strength class from C8/10 to C12/15;
- Normal strength concrete (NSC): strength class from C16/20 to C45/55;
- High strength concrete (HPC): strength class from C50/60 to C60/75;
- Ultra-high strength concrete (HSC): strength class from C70/85 to C100/120.

Within this collection, taking the classification of compressive strength related to "normal concrete" defined by standards DIN EN 206-1:2017-1 [44] and DIN 1045-2:2008 [58] as a reference, EPDs relating to concretes with a compressive strength value between C20/25 to C45/55 were selected.

Focusing on "normal concrete", the distribution of the environmental impacts for the selected indicators (GWP, ADPe, ADPf, AP, EP, ODP, POCP) of the ten concrete products selected by IBU are presented in Table 7.

Concerning the classification shown in Table 7, the sixth and tenth EPDs are excluded. The reason for the exclusion of EPD 6 is the lack of information on the management of the LCA, making it unsuitable for comparison with the rest of the selected products, while for EPD 10 it is the impossibility of fitting within the C20 to C55 range (see Table 6).

**Table 7.** Environmental impacts of concrete EPDs analysed, based on the same product category rules (PCR). The grey line represents the EPD excluded due to the non-consistent value for comparability purpose.

| N° | Impact Indicators | | | | | | |
|---|---|---|---|---|---|---|---|
| | GWP | ODP | AP | EP | POCP | ADPe | ADP$_f$ |
| | kg CO$_2$-eq. | kg CFC11-eq. | kg SO$_2$-eq. | kg (PO4)$^3$-eq. | kg ethene-eq. | kg Sb-Eq. | MJ |
| EPD1 | $1.78 \times 10^2$ | $4.79 \times 10^{-8}$ | $2.61 \times 10^{-1}$ | $4.98 \times 10^{-2}$ | $2.05 \times 10^{-2}$ | $6.08 \times 10^{-4}$ | $8.19 \times 10^2$ |
| EPD2 | $1.97 \times 10^2$ | $5.36 \times 10^{-8}$ | $2.87 \times 10^{-1}$ | $5.35 \times 10^{-2}$ | $2.30 \times 10^{-2}$ | $6.43 \times 10^{-4}$ | $9.00 \times 10^2$ |
| EPD3 | $2.20 \times 10^2$ | $1.69 \times 10^{-6}$ | $3.64 \times 10^{-1}$ | $3.47 \times 10^{-2}$ | $1.16 \times 10^{-1}$ | $4.40 \times 10^{-4}$ | $11.2 \times 10^2$ |
| EPD4 | $2.19 \times 10^2$ | $5.97 \times 10^{-8}$ | $3.17 \times 10^{-1}$ | $5.91 \times 10^{-2}$ | $2.58 \times 10^{-2}$ | $7.11 \times 10^{-4}$ | $9.97 \times 10^2$ |
| EPD5 | $2.44 \times 10^2$ | $6.81 \times 10^{-8}$ | $3.48 \times 10^1$ | $6.55 \times 10^2$ | $2.98 \times 10^2$ | $8.22 \times 10^{-4}$ | $1.08 \times 10^3$ |
| EPD6 | $3.73 \times 10^2$ | $1.72 \times 10^{-8}$ | $9.02 \times 10^{-1}$ | $1.22 \times 10^{-1}$ | $8.84 \times 10^{-2}$ | $3.51 \times 10^{-4}$ | $2.23 \times 10^3$ |
| EPD7 | $2.86 \times 10^2$ | $7.72 \times 10^{-8}$ | $4.06 \times 10^1$ | $8.10 \times 10^2$ | $3.51 \times 10^2$ | $1.02 \times 10^{-3}$ | $1.36 \times 10^3$ |
| EPD8 | $2.54 \times 10^2$ | $1.17 \times 10^{-8}$ | $6.72 \times 10^{-1}$ | $8.83 \times 10^{-2}$ | $6.46 \times 10^{-2}$ | $2.20 \times 10^{-4}$ | $1.57 \times 10^3$ |
| EPD9 | $3.00 \times 10^2$ | $8.40 \times 10^{-8}$ | $4.22 \times 10^1$ | $8.37 \times 10^2$ | $3.79 \times 10^2$ | $1.07 \times 10^{-3}$ | $1.36 \times 10^3$ |
| EPD10 | $4.66 \times 10^2$ | $9.39 \times 10^{-8}$ | $1.01$ | $1.40 \times 10^{-1}$ | $8.42 \times 10^{-2}$ | $5.08 \times 10^{-4}$ | $2.23 \times 10^3$ |

　　　The comparability among the compressive strength classes within the eight EPDs selected reveals a broad variability of the impact indicators in nearly all the nine indicators (Figure 6). In particular, whereas the variance of the global warming potential has a fairly linear trend and proportional growth from low to high compressive strength between the range of 10.7 to 17%, in the other indicators, especially in ODP, AP, POCP and ADPe, the impact values are highly variable. The comparison above showed moreover that the values of some indicators are not aligned with each other: e.g., the ODP in EPD3, the ADPe in EPD 8.

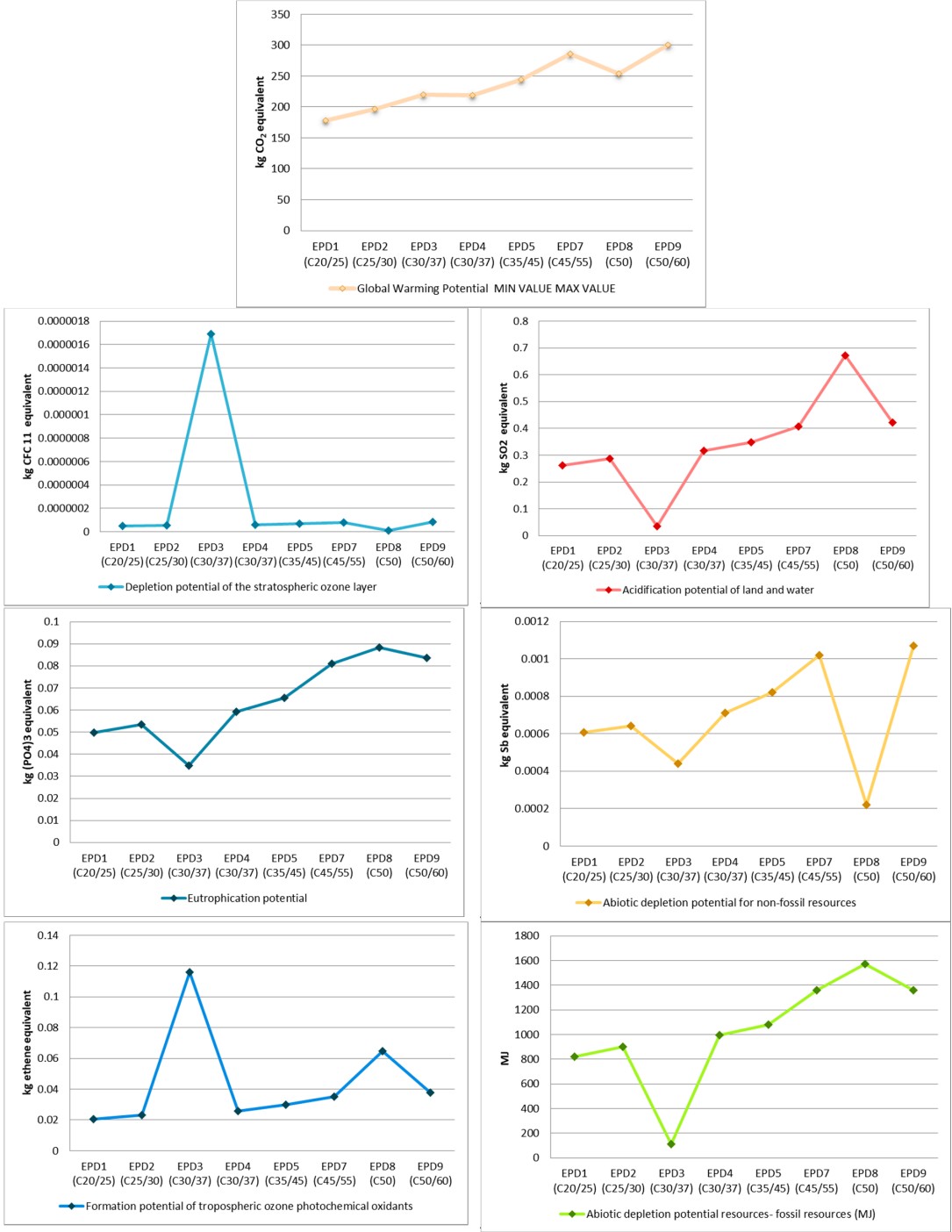

**Figure 6.** Variability of the impact indicators within the eight EPDs selected.

Therefore, once the classification shown in Table 7 was defined, and excluding the values that do not align with each other (EPD3 and EPD 8), the impact indicators were compared with the aim of evaluating the variability among them in two levels:

- for larger strength classes, namely when the solutions are established in the initial design stage (from LOD 100–200 to 300);
- and for contiguous strength classes, usable when the materials/products chosen are almost defined (e.g., from LOD 350 to 400).

2.2.4. Step 4: Determination of the Range of Values for Compressive Strength Grouping

Environmental Safety Factor in the Early Design Stage

With the intention of identifying a classification of concrete that can be used in the early design stage, the concrete has been divided into strength value groups according to its application in the building. In particular, it has been classified into strength groups (minimum and maximum range value), considering the principal building types (apartment house, administrative house, etc.) and taking into consideration the principal structural elements making up a concrete structure (foundation, floor structure, pillars) (Table 8).

**Table 8.** Classification of the compressive strength of a concrete structure with minimum and maximum values, based on the type of building and structural element.

| Building Type | Structural Elements | Compressive Strength | |
| :---: | :---: | :---: | :---: |
| | | **Min Value** | **Max Value** |
| **Apartment/House** | Floor structure/Foundation | C20/25 | C25/30 |
| | Pillars | C25/30 | C30/37 |
| | **RANGE** | **C20/25–C30/37** | |
| **Administrative Building** | Floor structure/Foundation | C30/37 | C45/55 |
| | Pillars | | |
| | **RANGE** | **C30/37–C45/55** | |
| **Industrial Building** | Floor structure/Foundation | C20/25 | C35/45 |
| | Pillars | C25/30 | C30/37 |
| | **RANGE** | **C20/25–C30/37** | |
| **Hospital** | Floor structure/Foundation | C35/45 | C50/60 |
| | Pillars | | |
| | **RANGE** | **C35/45–C50/60** | |

Environmental Safety Factor for Contiguous LODs

The definition of safety factors (SF) within contiguous compressive strength classes is presented in Table 9. The SF can be used by the designer when the commercial name of a product, code and manufacturer make are not known, hence the choice of a specific EPD.

**Table 9.** Safety factors (%) of concrete for seven environmental indicators available for bordering LODs.

| N° | GWP | ODP | AP | EP | POCP | ADPe | ADPf |
|---|---|---|---|---|---|---|---|
| | kg CO2-Eq. | kg CFC11-Eq. | kg SO$_2$-Eq. | kg (PO$_4$)$_3$-Eq. | kg Ethene-Eq. | kg Sb-Eq. | MJ |
| C20/25–C25/30 | 10.7% | 11.9% | 10.0% | 7.4% | 12.2% | 5.8% | 9.9% |
| C25/30–C30/37 | 11.2–11.7% | 11.4% | 10.5% | 10.5% | 12.2% | 10.6% | 10.8% |
| C30/37–C35/45 | 10.9–11.4% | 14.1% | 9.8% | 10.8% | 15.5% | 15.6% | 8.3% |
| C35/45–C45/55 | 17.2% | 13.4% | 16.7% | 23.7% | 17.8% | 24.1% | 25.9% |

## 3. Results and Discussion

The results shown in Figure 7 demonstrate that within contiguous strength classes of concrete, the values of the environmental impacts vary with values below 12.2% for the range classes C20/25 to C30/37, below 15.6% for C30/35–C35/45 and under 25.9% for high class C35/45–C45/55. As shown below (Table 9), each environmental impact category has a different percentage deviation.

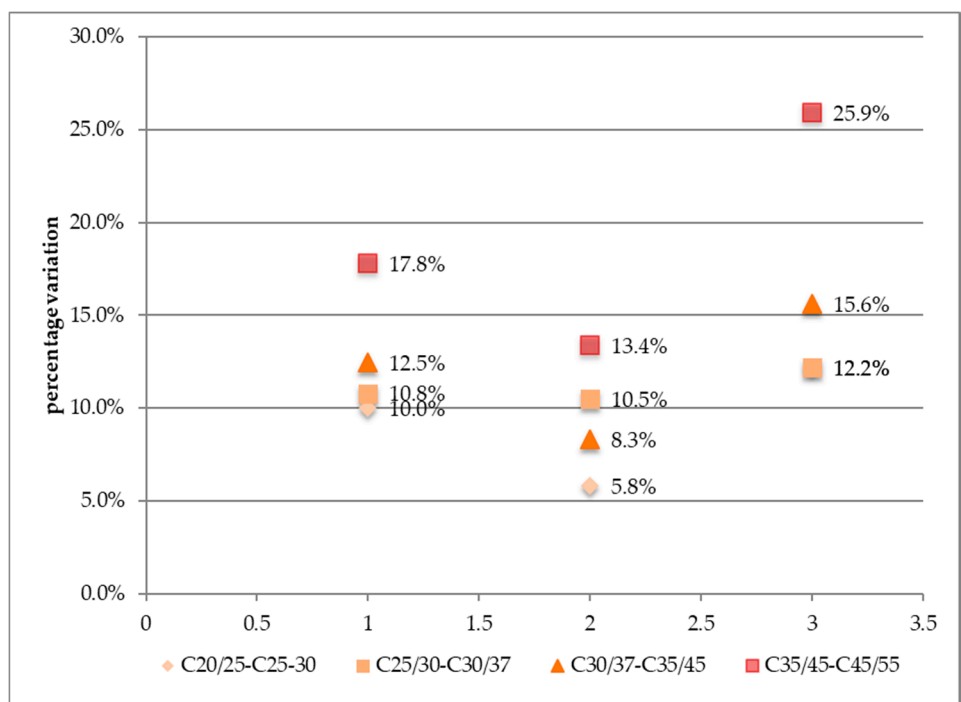

**Figure 7.** Percentage variation of the impact indicators within contiguous strength classes.

The comparison of concretes classified by type of building and structural elements, as proposed in Table 8, reveals that while considering a broader range of the compressive strength of concrete, the impact values fluctuating between the percentage values are not too dissimilar from those recorded in the more limited strength groups considered above (Figure 7). In particular, the values can vary by even up to 20–25% for two groups (C20/25 to 30/37 and C35/45 to 50/60) and only up to 30% for one group (C30/37 to 45/55). In contrast, a single factor that includes the variability between all ranges of "ordinary concrete" (from C20/25 to C50/60) increases the uncertainty of the assessment, reaching a percentage of up to 85% (Figure 8). The graph in Figure 9 depicts in greater detail the percentage variability of the concrete groups for each of the seven impact categories analysed (GWP, ODP, AP, EP, POCP, ADPe, ADPf). Figure 10 represents the variability of the results considering the concrete compressive strength classes of the selected EPDs. The graphics show that then highest variabilities are in the EDPs of the concrete compressive strength C45 for almost all the impact categories considered (GWP, ODP, EP, POCP, ADPe, ADPf).

The results shown in Table 9 and Figure 6 demonstrate that the steps to select the concrete based on the values extracted from EPDs belonging to a delimited range of structural performance can reduce the possible uncertainties and error factor to around 20% for the GWP, ODP, AP, EP, POCP, ADPe, ADPf impact categories in the design stages. These percentages rose slightly in the broader range concerning the classification of concrete considering the building typologies and their principal structural elements (Table 8).

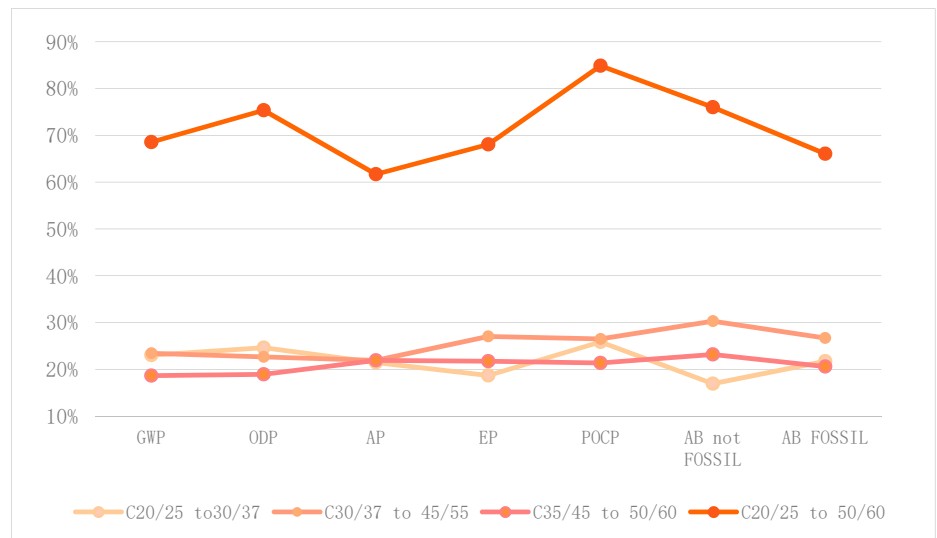

**Figure 8.** Percentage variation of environmental indicators within ordinary concrete.

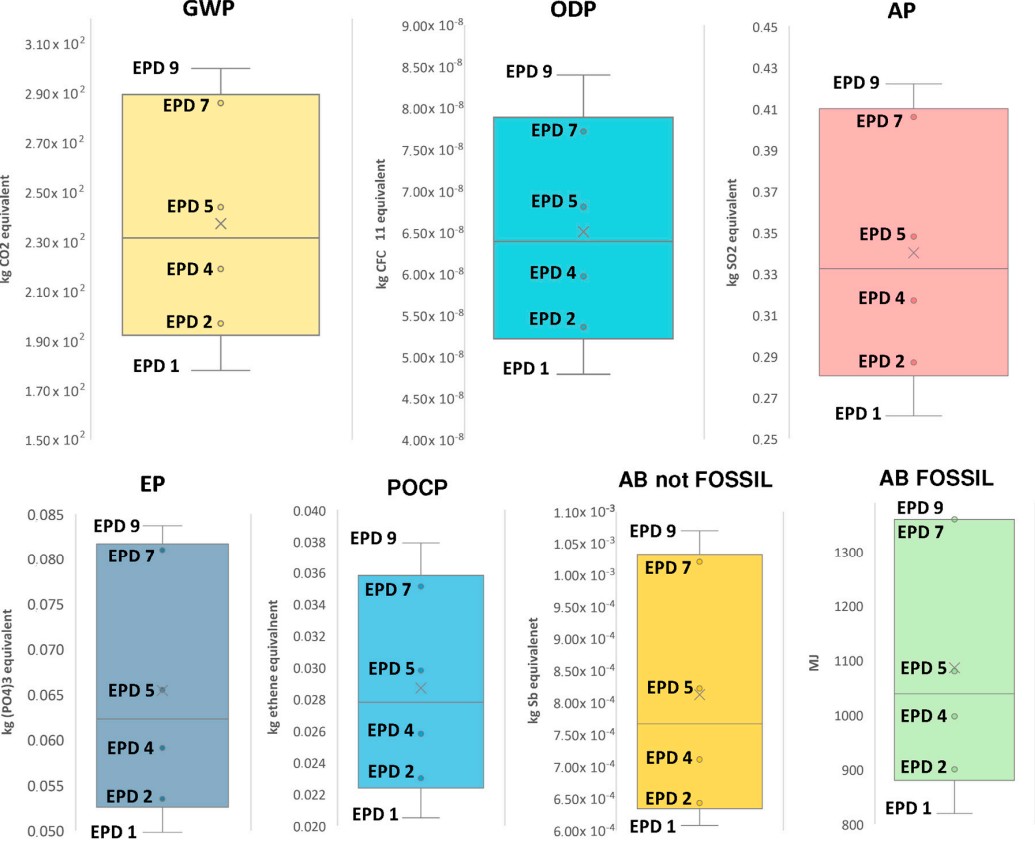

**Figure 9.** Variability of the selected EPD indicators for each of the seven environmental impact categories (GWP, ODP, AP, EP, POCP, ADPe, ADPf).

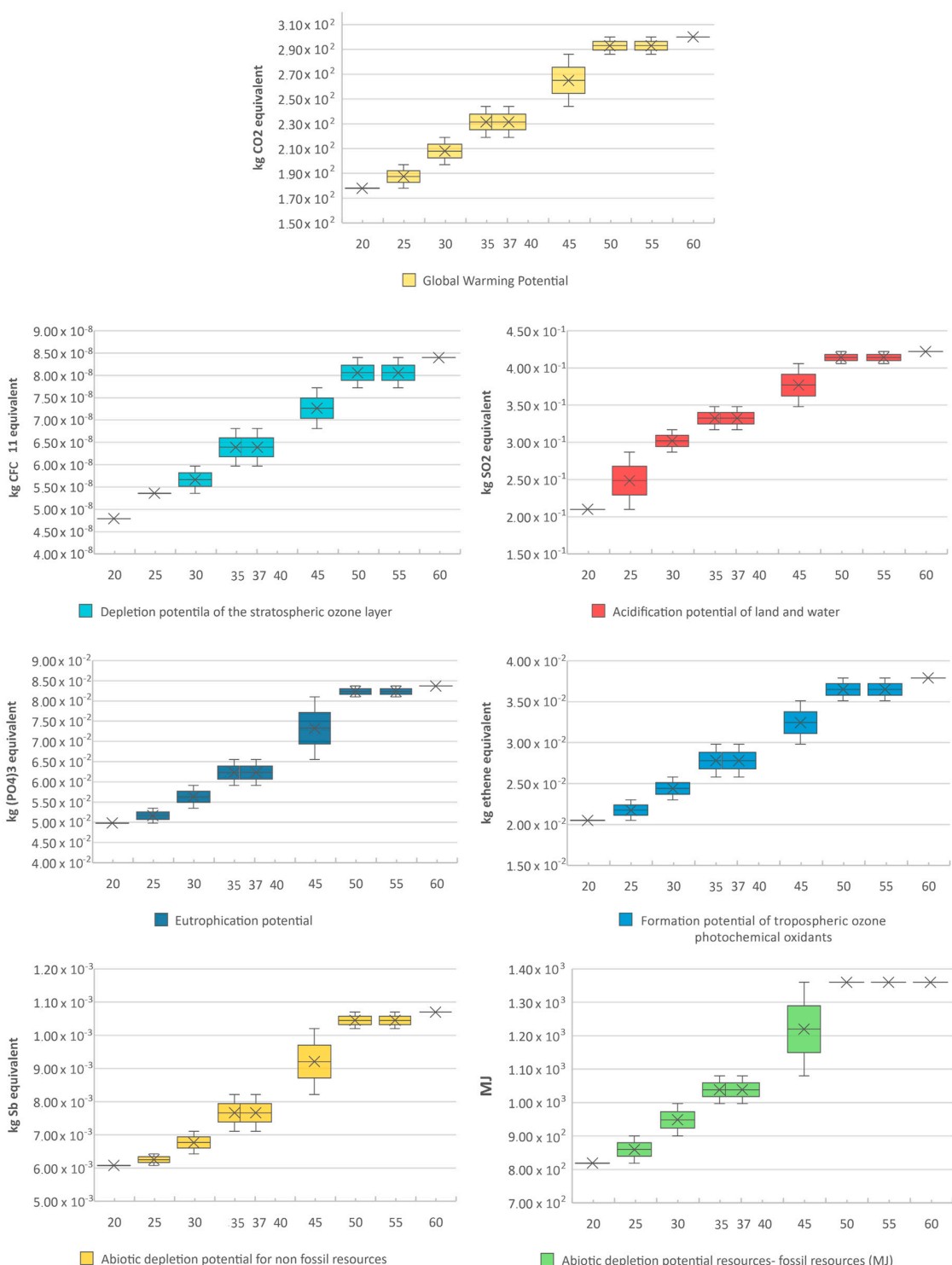

**Figure 10.** Variability of the results of the seven environmental impact categories (GWP, ODP, AP, EP, POCP, ADPe, ADPf) considering the concrete compressive strength classes of the selected EPDs.

The last step of the method proposed by this research consists of integrating the set of safety factors defined above in the BIM objects that compose the BIM model, assuming the workflow method proposed in Figure 11. Details of how each safety factor group can be integrated into BIM throughout the three main design processes, that is, early (LOD 100–200), intermediate (LOD 300–350) and detailed (above LOD 350), are given in Figure 12.

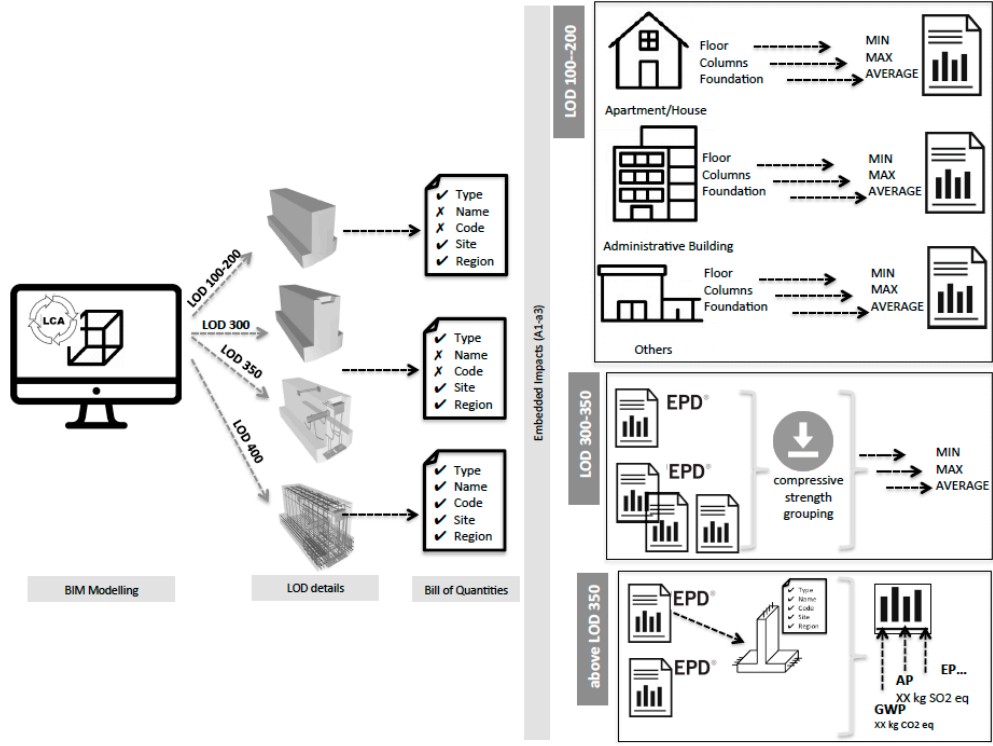

**Figure 11.** Integration into building information modelling (BIM): schematic workflow of the proposed method.

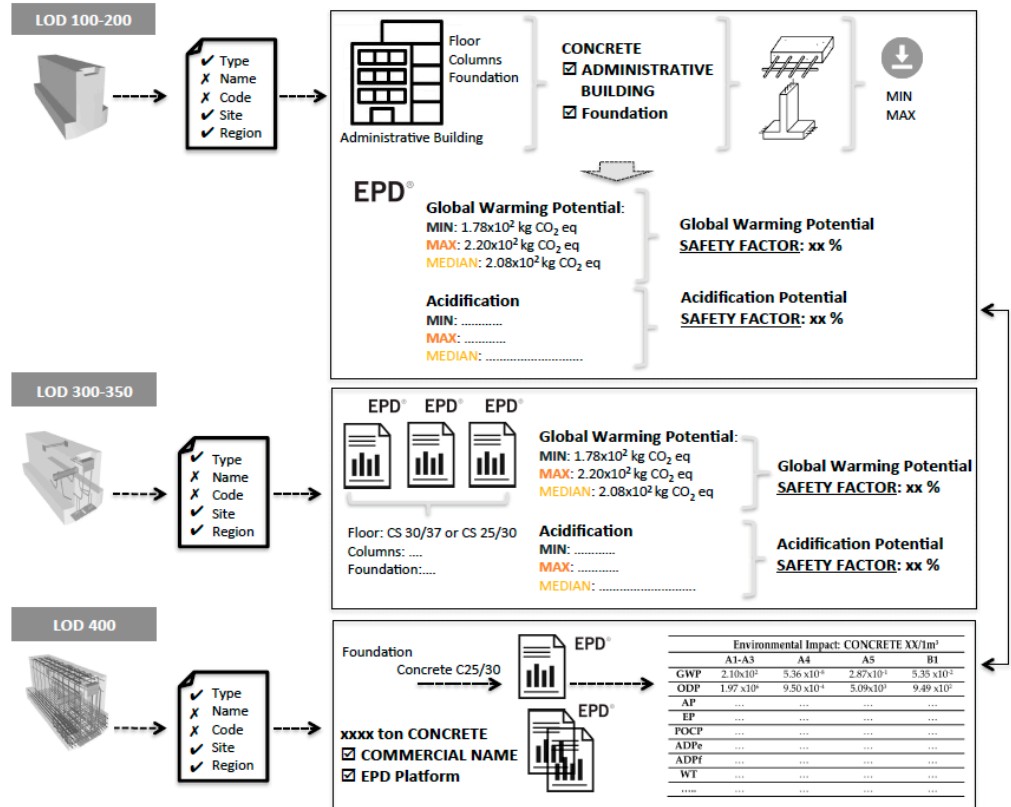

**Figure 12.** Integration of the safety factor in the early (LOD 100–200), intermediate (LOD 300–350) and detailed (above LOD 350) design stages.

The method presented combines the double definition of safety factors, one related to the mechanical performance (compressive strength) of the material and the other related to the environmental performance. Thus, the obtained results demonstrate that the definition of embodied impacts with accurate values based on EPDs should consider not only the environmental characteristics of the materials but also their performance, technical characteristics, PCR, PO, etc., to perform within a safe and reliable range of values.

## 4. Conclusions

The study carried out shows how in the current state, the use of EPD in the different LOD of BIM is not directly feasible but requires the information on environmental indicators and the performance of the materials and products to be treated appropriately before being applied.

The adoption of product-specific environmental data in the form of EPDs as a data source in LCA is not new. In green building rating systems (GBRSs), such as the English BREEAM [59] or the German DGNB, and more recently the level(s) developed by the European Commission [60], a voluntary framework for sustainability reports on buildings has been allowed for some time now. Besides, DGNB provides that, if EPDs do not completely match the specifications of the materials and components of the building, a safety margin of 10% should be applied to the LCA results to take into account possible deviations in the data provided by the EPDs. This represents a single safety value that does not take account of the distinction among environmental indicators or the typology of product. Regarding existing work to integrate the environmental information of EPDs in BIM, previous research has identified that they can only be included in higher LOD objects (LOD 400 and LOD 500) [61]. As regards the use of EPD as a data source in LCA instead of generic data, the literature has investigated this topic [38], concluding that EPDs are a reliable LCA data source, in accordance with the ILCD Guide. Of course, this approach has difficulty in finding application at low LODs, given that these levels do not include information on the brands or manufacturers of chosen products and the variability among LOD can be influenced by the total bill of materials quantities of the building. As shown by different authors [12–15], the early design stage has a great influence on the achievement of improved environmental performance.

This study proposes a method that helps us to obtain accurate and consistent LCA results, from the use of EPDs integrated into the building design stages in BIM. It demonstrates that the proposed steps reduce the range of possible errors or data inconsistency and help designers during decision-making. It uses the EPD as a tool to help with the selection of appropriate products and materials. The method and case study verification demonstrate that coherent results can be obtained from low to high BIM elements in LOD, without mixing different data granularity (generic and specific EPD) in conducting an LCA in BIM.

A case study validation based on a structural material like concrete demonstrates the possible application of the EPD in decision-making in the BIM methodology design process. The results obtained show the minimum, maximum and median values for the selected environmental impact categories that can be used at early design stages, which matches the EPD of the possible range of products that will be used at the detailed design stage.

The detected limitations of this method relate to the reduced number of considered EPDs. In fact, a lack of harmonized PCRs and non-homogenous formats of EPD schemes among programme operators has restricted the comparability of the existing environmental product declarations. Hence, a higher number of EPDs can better justify the use of the proposed method. This latter aspect is in line with the conclusions reached by Del Borghi and al. [18], which, referring to an earlier mapping [62], have also recently highlighted how the numerous existing PCRs and EPDs determine significant differences both as regards programmes and sectors, and PCR development. Product category rules are preconditions that ensure neutrality and the credibility of quantitative environmental information presented by EPDs. Therefore, in accordance with guidance documents released by the European Commission and the PCR Guidance Development Initiative, the authors recommend that operators

establish a common categorization system for specific sectors [62]. For all the above reasons and considering the key issue that the requirement on sustainability (especially the sustainable use of natural resources in the Basic Requirements For Construction Works 7) has in the Construction Products Regulation (CPR, EU No. 305/2011 [63]) [64], it is expected that the numbers of EPDs could increase in the future.

Moreover, the study was limited to embodied impacts from "cradle to gate" (including A1–A3 LCA modules). Future works should consider integrating A4 to D LCA modules of information, and consider how to deal with the environmental impact information in the BIM methodology workflow, also considering the aspects included in the progress standard on the use of EPD in BIM ISO/CD 22057 [65].

Even the classification of concrete presented in Table 8 needs to be refined and enhanced, as it was determined by the practical experience of the authors and lacks adequate literature references. The principal scope of the submitted classification was to propose an approach to the method and explain its development.

Another limitation of this study is that the scope of the method was focused on a specific building material (concrete) and its specific characteristics related to its performance in the building (structure). Future research will consider other types of materials that could be relevant for the total environmental impact produced by buildings, such as those making up the envelope (windows, walls and roofs) and interior walls, among others.

**Author Contributions:** Conceptualization, methodology, investigation, and data curation, E.P.; formal analysis, E.P. and B.S.-V.; project administration, E.P., B.S.-V., C.L. and M.T.; resources, E.P. and B.S.-V.; supervision, E.P. and B.S.-V.; validation, E.P., M.T., B.S. and C.L.; visualization, E.P. and B.S.-V.; writing-original draft preparation, E.P. and B.S.-V.; writing-review and editing, E.P., M.T., B.S.-V. and C.L.; funding acquisition, M.T. and C.L. All authors have read and agreed to the published version of the manuscript.

**Funding:** This research received no external funding.

**Acknowledgments:** The authors B.S.-V. and C.L. thank the Spanish Ministry of Science, Innovation and Universities, which partially supported the project entitled "Development of a unified tool for the quantification and reduction of environmental, social and economic impacts of life cycle buildings in Building Information Modelling platforms (BIM)" (Ref. BIA2017-84830-R).

**Conflicts of Interest:** The authors declare no conflict of interest.

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
