# Peer review of "How to Obtain Accurate Environmental Impacts at Early Design Stages in BIM When Using Environmental Product Declaration. A Method to Support Decision-Making"

_sustainability, doi:10.3390/su12176927_

Round 1
Reviewer 1 Report
Dear Authors,
According to the Construction Products Regulation, it is necessary to consider construction objects comprehensively while taking into account all the basic requirements. While the first six basic requirements have been widely present for years in the requirements, that construction products are subjected to conformity assessment before being placed on the market, the seventh basic requirement for sustainable development is still absent from the mandatory requirements. That undoubtedly has to change. This legal status affects the number of EPDs that could be available for and users and be used e.g. in BIM.
The paper is of high importance in the scientific field of sustainable design and the paper is very interesting. Moreover, the problem and methodology are described with sufficient accuracy taking into account all the flaws when comparing different EPDs obtained from different Program Operators.
Reviewer questions/comments:
Line 385. The question is, why EPD 7 was excluded from further verification as from Figures 6 it one can observe that from investigated impact indicators besides EPD 3, EPD 8 deviates the most in almost all of the categories investigated?
It could be interesting to see the dependence of investigated impact indicators in the function of concrete compressive strength. I could be depicted as point+/- deviation (if more points exist for certain concrete compressive strength) graph with trend line evaluated. Then the mean values with a certain level of confidence interval could be used as a reference to evaluate the "safety" factor (a reviewer suggestion to take under consideration).
Line 405. Preferably, table 9 could be placed before table 8 because there the compressive strength value ranges are not defined.
As authors mention in the conclusion that "the method was focused on a specific building material (concrete) and its specific characteristics related to its performance in the building (structure)." It would be great to consider External Thermal Insulations System with different insulation materials taking into account their thermal properties.
According to language, some minor spellchecks are required.
Best Regards,
Reviewer
Author Response
Reviewer 1 Dear Authors, According to the Construction Products Regulation, it is necessary to consider construction objects comprehensively while taking into account all the basic requirements. While the first six basic requirements have been widely present for years in the requirements, that construction products are subjected to conformity assessment before being placed on the market, the seventh basic requirement for sustainable development is still absent from the mandatory requirements. That undoubtedly has to change. This legal status affects the number of EPDs that could be available for and users and be used e.g. in BIM. Thank you for this essential reference about CPR that we have neglected to insert (see line 544). The paper is of high importance in the scientific field of sustainable design and the paper is very interesting. Moreover, the problem and methodology are described with sufficient accuracy taking into account all the flaws when comparing different EPDs obtained from different Program Operators. Reviewer questions/comments: Line 385. The question is, why EPD 7 was excluded from further verification as from Figures 6 it one can observe that from investigated impact indicators besides EPD 3, EPD 8 deviates the most in almost all of the categories investigated? Than you for the essential observation, it is not excluded EPD 7 but 8, is a misprint, we have changed it in the text. It could be interesting to see the dependence of investigated impact indicators in the function of concrete compressive strength. I could be depicted as point+/- deviation (if more points exist for certain concrete compressive strength) graph with trend line evaluated. Then the mean values with a certain level of confidence interval could be used as a reference to evaluate the "safety" factor (a reviewer suggestion to take under consideration). Thank you for the suggestion, we have included Figure 9 (page 17), which represents the variability of the impact indicators in the function of concrete compressive strength of the selected EPDs. Line 405. Preferably, table 9 could be placed before table 8 because there the compressive strength value ranges are not defined. Thank you for the useful suggestions. We have replaced the tables. As authors mention in the conclusion that "the method was focused on a specific building material (concrete) and its specific characteristics related to its performance in the building (structure)." It would be great to consider External Thermal Insulations System with different insulation materials taking into account their thermal properties. Thank you for the interesting proposal to analyse the external thermal insulations system, unfortunately, the data concerning the impact indicators of the insulation layer are not currently included in the EPDs analyzed. In fact, the EPDs considered are referred to a concrete material and not a complete system or building envelope. According to language, some minor spellchecks are required. We have checked the test by native English speakers. Best Regards, Reviewer

Reviewer 2 Report
According to the authors:
- a higher number of EPDs can better justify the use of the proposed method;
- the classification of concrete presented in Table 9 needs to be refined and enhanced, as it lacks adequate literature references;
- another limitation of the study is that the scope of the method was focused on a specific building material (concrete) and its specific characteristics related to its performance in the building (structure).
Author Response
Reviewer 2
Comments and Suggestions for Authors
According to the authors:
a higher number of EPDs can better justify the use of the proposed method;
Thank you for the suggestion, we have added it (line 492).
the classification of concrete presented in Table 9 needs to be refined and enhanced, as it lacks adequate literature references;
Thank you for the suggestion, we have added it (line 505).
another limitation of the study is that the scope of the method was focused on a specific building material (concrete) and its specific characteristics related to its performance in the building (structure).
Thank you for the suggestion, we have added it (line 556).